# Kinematics Governing Mechanotransduction in the Sensory Hair of the *Venus flytrap*

**DOI:** 10.3390/ijms22010280

**Published:** 2020-12-30

**Authors:** Eashan Saikia, Nino F. Läubli, Jan T. Burri, Markus Rüggeberg, Christian M. Schlepütz, Hannes Vogler, Ingo Burgert, Hans J. Herrmann, Bradley J. Nelson, Ueli Grossniklaus, Falk K. Wittel

**Affiliations:** 1Institute for Building Materials, Swiss Federal Institute of Technology Zurich (ETH Zürich), 8093 Zurich, Switzerland; esaikia@ethz.ch (E.S.); mrueggeberg@ethz.ch (M.R.); iburgert@ethz.ch (I.B.); hans@ifb.baug.ethz.ch (H.J.H.); 2Department of Mechanical and Process Engineering, ETH Zurich, 8092 Zurich, Switzerland; laeublin@student.ethz.ch (N.F.L.); jan.burri@gmx.net (J.T.B.); bnelson@ethz.ch (B.J.N.); 3Swiss Federal Laboratories for Materials Science and Technology—Empa, Cellulose & Wood Materials Laboratory, 8600 Dubendorf, Switzerland; 4Swiss Light Source, Paul Scherrer Institute, Forschungsstrasse 111, 5232 Villigen PSI, Switzerland; christian.schlepuetz@psi.ch; 5Department of Plant and Microbial Biology & Zurich-Basel Plant Science Center, University of Zurich, 8008 Zurich, Switzerland; hannes.vogler@botinst.uzh.ch (H.V.); grossnik@botinst.uzh.ch (U.G.)

**Keywords:** *Dionaea muscipula*, *Venus flytrap*, plant biomechanics, mechanotransduction, multi-scale modelling, turgor pressure, sensory hair

## Abstract

Insects fall prey to the *Venus flytrap* (*Dionaea muscipula*) when they touch the sensory hairs located on the flytrap lobes, causing sudden trap closure. The mechanical stimulus imparted by the touch produces an electrical response in the sensory cells of the trigger hair. These cells are found in a constriction near the hair base, where a notch appears around the hair’s periphery. There are mechanosensitive ion channels (MSCs) in the sensory cells that open due to a change in membrane tension; however, the kinematics behind this process is unclear. In this study, we investigate how the stimulus acts on the sensory cells by building a multi-scale hair model, using morphometric data obtained from μ-CT scans. We simulated a single-touch stimulus and evaluated the resulting cell wall stretch. Interestingly, the model showed that high stretch values are diverted away from the notch periphery and, instead, localized in the interior regions of the cell wall. We repeated our simulations for different cell shape variants to elucidate how the morphology influences the location of these high-stretch regions. Our results suggest that there is likely a higher mechanotransduction activity in these ’hotspots’, which may provide new insights into the arrangement and functioning of MSCs in the flytrap.

**Dataset:** 10.3929/ethz-b-000448954

## 1. Introduction

The *Venus flytrap* (*Dionaea muscipula* Ellis) has fascinated scientists since the time of Darwin [1] and, over the past century, studies have shed light on different aspects of trap closure, such as the snap-buckling instability [2], signalling pathways in the flytrap [3], and the electrical response to a mechanical stimulus [4]. In 1834, Curtis highlighted the significance of sensory hairs on the lobes for transforming the mechanical stimulus [5]. Since then, the sensory hair has sought the curiosity of many scientists. However, the basic mechanism behind mechanotransduction within the hair remains elusive at best. In this work, we explore how the mechanical stimulus acts on the sensory cells present in the constriction near the base of the hair, which are known to generate an electrical response [6].

Typically, six sensory hairs are found on the upper epidermis of each flytrap, three on each lobe. When an insect touches these sensory hairs, the lobes close if sufficiently stimulated. One of the first comprehensive microscopic investigations of the sensory hair’s anatomy was made by Haberlandt in 1906, which included a short description of the sensory cells [7]. In a longitudinal hair section, he portrays a sharp notch on the peripheral side of the sensory cells with strongly reduced cell wall thickness. In 1971, Transmission Electron Microscopy (TEM) images of the sensory hair [8] showed that cell wall thickness varies across the sensory cells. This could imply that specific regions of the cell wall may deform to different extents upon stimulation.

Mechanical tests on sensory hairs were able to quantify the stimulus thresholds needed to elicit action potentials (APs) and, furthermore, a loss of sensitivity at high stimulation frequencies was reported [9]. Recently, with the help of an electromechanical model, it was shown that a charge-dependent rule can explain mechanotransduction in the flytrap’s sensory hair [10]. Subsequent experiments confirmed that the number of stimulations needed to close the trap depends on the stimulus parameters, namely the angular velocity and the angular displacement of the sensory hair. These findings describe mechanotransduction in the flytrap at the macroscopic level. What remains to be established is the kinematic link between sensory hair deflection and sensory cell deformation. In a recent report, the calcium signaling cascade was visualized using transgenic *Venus flytraps*, whereby a stimulus initiated an electrochemical response near the hair base, which then spread to the flytrap’s lobes [11]. Previous studies suggest the existence of mechanosensitive ion channels (MSCs) in the sensory cells [3]. MSCs respond to a change in membrane tension caused by stretching [12,13] or contraction of the plasma membrane [14]. However, evaluating membrane stretch in the sensory hairs is experimentally challenging. In vivo stimulation of sensory hairs can lead to sudden trap closure, thus making it difficult to repeat the experiment. Alternately, tests on excised hairs lead to a loss of turgor pressure, which significantly alters the hair’s natural behavior.

Computational approaches have hardly been used to bypass the challenges associated with the characterization of *Venus flytraps*. Here, we developed Finite Element Method (FEM)-based models of the sensory hair and its tissues comprised of fluid-filled cells. To build the models, we obtained the 3D geometric profile of the sensory hair and the internal cellular morphology from a set of μCT images. Then, we simulated a single-touch stimulus by deflecting the sensory hair up to a predefined angular displacement (α). We selected the range of α such that a single stimulus can initiate trap closure as previously observed in our experiments [10]. As a result of the stimulus, we obtained the cell wall deformation as an output, from which we calculated the stretch produced on the cell wall. Contrary to our expectations, we found that the highest cell wall stretch is produced in regions away from the location of the notch. To further investigate this finding, we considered two additional cell shapes, a notch-less cell and a cell with uniform wall thickness, and repeated the simulations. Our findings show that cell morphology strongly affects the location of the highly stretched regions. It is probable that these regions are mechanotransduction ‘hotspots’ where MSCs may be densely arranged. These findings point towards an adapted cell geometry, whereby an optimized cell morphology directs the mechanical stimulus into certain regions of higher sensitivity.

## 2. Materials and Methods

The study combines different experiments with numerical modelling to simulate a mechanical stimulus on the flytrap’s sensory hair. Firstly, we made μ-CT scans of the hair and gathered the relevant morphometric data. With this information, we built multi-scale models of the sensory hair. To calibrate the material parameters, we performed force-deflection tests on the sensory hair and compared the experimental stiffness measurements with those obtained from our simulations. Finally, we deflected the sensory hair model to simulate a single-touch stimulus and computed the resulting stretch in the sensory cells.

### 2.1. Anatomy of the Venus flytrap’s Sensory Hair

*Venus flytraps* were taken from a population of 100 plants kept in the greenhouse of the Department of Plant and Microbial Biology of the University of Zurich for the morphological study of their sensory hairs and subsequent force-deflection tests. These plants were originally grown from seeds donated by the Botanical Garden Zurich (https://www.bg.uzh.ch) in 2011. We used a mixture of 90% white peat (Zuercher Blumenboerse, Wangen, Switzerland) and 10% granulated clay (SERAMIS Pflanz-Glanulat, Westland Schweiz GmbH, Diesldorf, Switzerland) as substrate. The plants were kept in the greenhouse at 60% relative humidity and a temperature range of 18 ${}^{\circ }C$ to 23 ${}^{\circ }C$ during the day and 16 ${}^{\circ }C$ to 21 ${}^{\circ }C$ during the night. Plants were grown under normal daylight, morning and evening periods being extended by 400 W metal-halide lamps (PF400-S-h, Hugentobler Spezialleuchten AG, Weinfelden, Switzerland) to ensure a day-length of 16 h. The lamps were also turned on when the daylight was not sufficient.

There are three kinds of tissues identified in the sensory hair: the proximal podium, the constriction, and the distal lever. The podium is located near the base of the hair and connects it to the upper epidermis of the flytrap lobe (Figure 1a). The constriction is located above the podium and can be distinguished by the sharp notch present around its periphery. The distal lever is located above the constriction and has a pointed tip. Sensory cells with kidney-like shapes (light green color) are present in the constriction while the other tissues consist of elongated cylindrical cells (blue, yellow).

#### 2.1.1. Micro-CT Imaging and Data Post-Processing

The purpose of taking μ-CT scans of the sensory hair was to gather morphological data needed to build 3D multi-scale models of the hair, as described in Section 2.3. Each scan sample comprised of a circular lobe section (3 mm in diameter) containing in its center a single sensory hair, which was punched out from the trap using a biopsy needle (Figure 1c). The sample was fixed onto a sample holder with nail polish, which was also applied to seal the edges to prevent dehydration. 3D synchrotron X-ray CT data was acquired for a total of three hairs at the TOMCAT beamline X02DA of the Swiss Light Source facility at the Paul Scherrer Institute, Villigen, Switzerland [15]. The energy of the X-ray beam was chosen at 12 keV using a broad-band multilayer double-crystal monochromator. At this energy, we obtained a good image contrast while, at the same time, providing a reasonable degree of edge enhancement to exploit the partial coherence of the X-ray beam using a propagation-based phase contrast method and to avoid excessive beam damage imparted by the X-rays on the sample. X-ray projection images were converted to visible light by a 20 μm thick LuAG:Ce scintillator (Crytur, Turnov, Czech Republic), optically magnified by a factor of 20 with a dedicated microscope (Optique Peter, Lentilly, France) and recorded by a pco.edge 5.5 scientific CMOS camera (PCO, Kelheim, Germany) with a pixel size of 6.5 μm, which results in an effective pixel size of 0.325 μm when taking the magnification into account.

Tomographic scans consisted of 901 projections in the range [0${}^{\circ }$, 180${}^{\circ }$], resulting in angular increments of 0.2${}^{\circ }$. Each projection image has 2560×2160 pixels. The exposure time used for each projection was 50 ms, resulting in ≈45 s of total exposure time for each scan, which is sufficiently short to mitigate radiation effects while providing a good data quality. The collected radiographs were corrected for flat and dark field projections, which was then followed by the application of a Paganin phase filter [16]. The sinograms from these phase contrast projection images were reconstructed using the Gridrec algorithm [17] to obtain the final reconstructed 3D image of the individual hair sections. The image stack obtained from a single scan contained only a vertical region of the hair. Using the Image Processing Toolbox of MATLAB 2018b, the individually measured vertical regions were stitched together and blended to obtain the complete hair stack for each hair (see Appendix A). The complete stack was cropped, its contrast was adjusted, and 3D filters were applied to enhance anatomical features.

We found the average geometric profile of the sensory hair from μ-CT scans of three different hairs. Firstly, the 3D stacks of the hairs were rotated to align it in the longitudinal ‖ direction. Then, the area of the hair cross-sections (⊥ plane) in every CT slice was evaluated along the entire hair height. Circles with equivalent area were fitted to each cross-section and their corresponding radii *r* were stored. Finally, the geometry of the sensory hair was defined as a function of two dimensionless parameters $r^{*}=f\left(h^{*}\right)$ (see Figure 2a), where $h^{*}$ is the normalized hair height with respect to the average hair height, and $r^{*}$ corresponds to the normalized cross-sectional radius at a certain height. The first order derivative $f^{\prime }\left(h^{*}\right)$ of this profile was used to detect the gradients in the hair profile. In this way, we identified 6 characteristic regions of the hair on the basis of notable changes in radii of cross-sections, as [$r_{i}$, $h_{i}$] for i∈ [1, 6] and listed in Table 1.

From the μ-CT data, we measured the cellular features and parameterized them (see Figure 2b) into the cell wall thickness $T_{c}$, lumen diameter $D_{c}$, and the cell height $H_{c}$, whose values are listed in Table 2. Sensory cells have a distinct kidney-like shape, with a notch formed by additional thickening of the peripheral cell wall (Figure 2c). Measurements of the notch geometry were taken from (n = 8) images of the longitudinal section of sensory cells, which were then normalized with respect to the average cell height. Lastly, cross-sectional images of sensory cells (n = 51) at the middle (Y-Y), and at two more regions (X-X) located at one quarter and three-quarters of the total height $H_{s}$, respectively, were used to measure the wall thicknesses $T_{s}$ and the cell opening angle θ.

#### 2.1.2. Light Microscopy for Cell Wall Thickness Measurements

We measured the cell wall thicknesses under a microscope to calibrate the measurements taken from the post-processed CT data (see Section 2.1.1). For this purpose, sensory hairs were cut from three different lobes of a single plant and embedded in an aqueous solution of 50% (*v*/*v*) PEG 2000 (polyethylene glycol melted at 60 ${}^{\circ }C$), similar to the procedure in [18]. The hair samples were then transferred into individual containers with the PEG solution and were kept in the oven for three days at 57 ${}^{\circ }C$. Following this, pure PEG was added to refill the containers after the initial water had evaporated and the samples were reheated for 24 h to remove the remaining air within the solution. The cooled and hardened samples were then removed from the containers. With a rotational microtome, cross-sectional slices of the hair (2 μm thick) were cut from the lever and the podium tissues. The microtome slices were transferred onto a microscope slide, the PEG was dissolved with a drop of water, and the tissue was stained with 1% aqueous Safranine solution. Transmission light images were taken (see Figure 3b) using a Brightfield Microscope at 20×, 40×, and 100× magnification. From these images, cell wall thickness (n = 30) was measured for three slices of constriction tissue ($h_{5}<h<h_{3}$) and (n = 25) for three slices of the lever tissue ($h_{2}<h<h_{1}$).

### 2.2. Force-Deflection Tests on Sensory Hairs

We performed force-deflection tests on sensory hairs to find the moment-based hair stiffness. Afterwards, we compared this value with the corresponding stiffness values calculated via simulation to tune our model parameters. A total of 17 deflection tests were made on seven sensory hairs taken from four individual plants. The test set-up (seen in Figure 1b) included a MEMS-based force sensor (FT-S1000-LAT; FemtoTools AG, Buchs, Switzerland), which is integrated into a microbotics system to control its movement. The force sensor has a range of ±1000 μN with a standard deviation of 0.09 μN at 200 Hz, and it measures forces along the direction of motion. The force signal was recorded with a multifunction I/O device (NI USB-6003; National Instruments (NI), Austin, TX, USA). The deflection force sensor was mounted via a custom-made acrylic arm to a piezoelectric *xyz*-positioner (SLC-2475-S; SmarAct, Oldenburg, Germany) with a closed-loop resolution of 50 nm.

The two lobes of the trap, containing the target sensory hair, were constrained using metallic clamps. This ensured that the force sensor is not damaged in the event of a trap closure. Once constrained, the force sensor was guided into the flytrap with the help of two USB microscope cameras (DigiMicro Profi; DNT, Dietzenbach, Germany) with mutually perpendicular planes of view. The longitudinal axis of the sensory hair was in the plane of view of the first microscope, thereby allowing us to evaluate the hair height and the sensor contact height $h_{x}$. The second microscope contained the force sensor in its plane of view and was used to bring the probe into close proximity of the sensory hair. After this, the probe was advanced horizontally to deflect the hair by a distance Δx. The reaction force $F_{x}$ at the contact point was continuously measured and the resulting moment $M_{x}=F_{x}\times h_{x}$ was subsequently evaluated (see Figure 4b). Lastly, the moment-based stiffness *k* of the sensory hair, given by the initial linear slope of the curve $M_{x}$ vs. Δx, was calculated.

### 2.3. Multi-Scale Finite Element (FE) Model of the Sensory Hair

Using the morphometric data of the sensory hair and its internal cells, obtained from Section 2.1.1, we build a multi-scale model of the hair in order to simulate the force-deflection tests performed in Section 2.2. We followed a hierarchical approach, comprising of a macro model of the sensory hair (see Figure 5a) and micro models of the different parenchymatous tissues of the hair (see Figure 5b). All simulations are performed with geometric non-linearity in ABAQUS 2019a.

#### 2.3.1. Homogenization of the Tissue Micro Models

We built micro models (see Figure 5b) of the sensory hair tissues and calculated their coarse-grained material properties, which were given as input into the macro model in Section 2.3.2. The micro models of the lever (blue) and podium (yellow) consist of hexagonal cells and their geometry is parameterized by the lumen diameter $D_{c}$, height $H_{c}$, and cell wall thickness $T_{c}$ (See values in Table 3). Cell walls are modeled via quadratic shell elements S8R, based on the thick shell theory by Mindlin [19], and they are assumed to have an isotropic Young’s modulus $E_{cw}$, and Poisson’s ratio of 0.3. The assumption of isotropy is hereby based on measurements of the cellulose orientation distribution in the cell walls throughout the lever, using wide angle X-ray diffraction (see Section A.2). Note that a linear material is assumed for the cell wall because our focus is on the influence of geometric features of the sensory hair. Adjacent cell walls are merged into one shell element having the thickness $2T_{c}$ and the cell lumina are modeled as closed cavities. To obtain a representative behavior of the tissues, each micro model considers 19 cells placed next to each other in a single layer, and three such cell layers are stacked along the ‖ direction with randomly staggered cells.

Each numerical simulation consists of two static steps, wherein during the first step, a ‘fluid cavity pressure’ boundary condition fills the lumina with an incompressible fluid up to a predefined turgor pressure *P*. A planar symmetry boundary condition is applied on the bottom surface to seal the open ended cells, thus preventing any leakage of fluid. The ‘fluid cavity pressure’ condition is deactivated at the end of the first step to maintain a constant fluid volume later. In the second step, we constrained one surface of the model with a planar symmetry condition, to restrict any out-of-plane movement. The opposite surface is kinematically linked to a reference point (RP), which is then displaced up to 1% tensile and shear strains separately, and the reaction forces on the RP are given as an output. In the case of shear displacement, all DOF of the constrained surface are set to 0 by using the ‘Encastre’ boundary condition. The secant slope of the respective force-displacement curves is used to calculate the elastic and shear moduli in the ‖ and ⊥ directions. In this way, the 6 material parameters needed to populate an orthotropic stiffness tensor are found and the three Poisson’s ratios are measured. Following this, the orthotropic material tensor is reduced to a transversely-isotropic material tensor by taking the average of the values in the two transverse directions.

We performed a parametric study with input parameters ($100M\mathrm{Pa}\leq E_{cw}\leq 400M\mathrm{Pa}$) based on earlier studies on parenchymatous tissues [20] that estimated similar values for $E_{cw}$. We varied *P* between 0–0.3 MPa, a range was taken from previous experiments concerning *Mesembryanthemum* bladders, which exhibited a full turgor pressure between 0.3–0.4 MPa [21].

#### 2.3.2. Macro-Scale Numerical Hair Model and Deflection Simulations

We built a macro model of the sensory hair to simulate the deflection tests in Section 2.2 and to obtain the deformation of the sensory cell as an output. The total height of the hair model is 1500 μm but, to save on computational costs, we did not model the lever tissue present above the force sensor contact point in our model. The macro model of the sensory hair (Figure 5a) is a hybrid between a cellular model of fluid-filled sensory cells and solid models of the lever (blue) and podium (yellow), having the homogeneous material properties as calculated in Section 2.3.1. A cell wall layer of thickness $t_{a}=5\mu m$ surrounds the podium and the lever tissues to account for the additional thickening of the periphery observed in the μ-CT images (See Figure 3c). The cellular model considers 16 sensory cells with isotropic cell wall material (beige color), which are distributed evenly around the constriction (Figure 5c). A segmented view of the inner cell lumina (green) is also shown, with positions defined by their polar angle ϕ about the central height axis. The cells at the extreme left ($\phi =0^{\circ }$) and the extreme right ($\phi =180^{\circ }$) are modeled as half cells due to the intersection of the symmetry plane.

The original shape (Figure 5d(i)) of the sensory cell contains a sharp notch with varying cell wall thicknesses along the cell’s height. The thickest section ($T_{s,max}$) is in the middle and the thinner sections ($T_{s,min}$) are located at the top and the bottom. To study the influence of the geometric features, we designed two shape variants (Figure 5d(ii, iii)): a notchless sensory cell (N.C) lacking the notch and a sensory cell (U.C) with a constant cell wall thickness. The N.C also has varying wall thicknesses but the U.C lumen is modified such that the total cell wall volume of the two variants remains approximately equal. The sensory cell walls have the same material properties as those in the tissue micro models, with elastic moduli $E_{cw}$ and Poisson’s ratio = 0.3. The FEM mesh of the cell wall primarily consists of quadratic solid hexahedral elements with reduced integration (C3D20R) and a few quadratic tetrahedral elements (C3D10) in the narrow curved regions for better geometric approximation.

In the solid model, the base of the podium is fixed with an ’Encastre’ boundary condition and a ’planar symmetry’ boundary condition is imposed on the surfaces lying on the symmetry plane. The planar symmetry also seals the open lumina of the diametrically opposite half cells located at $\phi =0^{\circ },180^{\circ }$. In the first step of the static simulation, the sensory cells were inflated with an incompressible fluid up to a turgor pressure *P* = 0.2
MPa by using the ’fluid cavity pressure’ boundary condition. In the second static step, the distal lever is deflected horizontally along the symmetry plane up to Δx=300μm (Figure 1d). To deflect the sensory hair, we used a rigid body that contacts the lever at a height $h_{x}=987\mu m$, which is approximately at two-third of the total height of the sensory hair. The corresponding angular displacement α is given by the relation
(1)$$\alpha =\arctan\frac{\Delta x}{h_{x}},$$
with a maximum value of $\alpha =16.9^{\circ }$ at Δx=300μm. The reaction force $F_{x}$ at the point of contact is given as an output, from which we calculated the reaction moment $M_{x}=F_{x}\times h_{x}$. Thereafter, the moment-based stiffness $k_{num}$ is found by taking the initial linear slope of the $M_{x}$ vs. Δx curve. By comparing $k_{num}$ with the experimental stiffness values obtained in Section 2.2, we defined the appropriate set of input parameters *P* and $E_{cw}$, which can best represent the observed mechanical behavior of the sensory hair.

#### 2.3.3. Calculation of Mechanotransductive Properties

MSCs open due to a change in membrane tension when the membrane stretches or contracts [14]. Therefore, we calculated the stretch produced on the inner lumen surface (see Figure 5d) of the cell walls based on the hair deflection simulations in Section 2.3.2. We post-processed the simulation outputs using the ABAQUS-Python environment and MATLAB 2019a. Firstly, nodal outputs of the strain tensors were extracted from the mesh nodes lying on the inner lumen surface. For every element *n*, an averaged strain tensor was calculated by finding the arithmetic mean of all the strain tensors of the associated nodes and, which was then transformed into the local element coordinate system. Following this, the 3D strain tensors were reduced to a 2D strain tensor by assuming a plane-stress condition and the in-plane principal strain components $e_{1}^{n}$, $e_{2}^{n}$ were calculated. These strain components were used to compute the stretch $\lambda^{n}$ produced in every element given by
(2)$$\lambda^{n}=\left(1+e_{1}^{n}\right)\;\cdot \;\left(1+e_{2}^{n}\right).$$

Following this, we calculated two coarse-grained mechanotransductive properties for each sensory cell located at a polar angle ϕ, namely the cumulative area-normalized stretch $\delta_{\phi }$ and the sensitive area fraction $\rho_{\phi }$. Firstly, $\delta_{\phi }$ is calculated by using the relation
(3)$$\delta_{\phi }=\frac{\Sigma (\lambda^{n}\;\cdot \;A_{n})}{\Sigma A_{n}},$$
where $A_{n}$ is the surface area of the element *n* and normalization with the total area results in $\delta_{\phi }$, being a mesh-independent scalar value. With this, we were able to compare the cumulative stretches among the three cell shape variants described in Figure 5d. Next, we made the simple assumption that there could be threshold values of stretch $\lambda_{max}$ or contraction $\lambda_{min}$ on the inner lumen surface, above and below which MSCs may open. Here, we defined the second property as the sensitive area fraction $\rho_{\phi }$, which is is given by:(4)$$\rho_{\phi }=\frac{\Sigma (A_{n,\lambda^{n}})}{\Sigma A_{n}},$$
where $A_{n,\lambda^{n}}$ is the area of the element *n* in the mesh, which has a stretch $\lambda^{n}\geq \lambda_{max}$ or $\lambda^{n}\leq \lambda_{min}$. In order to compare different regions within the same cell, we divided the cell wall into three regions: top, middle, and bottom. The corresponding region-specific coarse-grained properties $\delta_{top,mid,bot}$ and $\rho_{top,mid,bot}$ were evaluated by using Equations (Equation 3) and (Equation 4).

## 3. Results

Our work builds on previous descriptions of the anatomy of sensory hairs, the earliest of which exist in the form of drawings of the sensory cells sketched by Haberlandt [7]. We extended these observations with new 3D μ-CT data and parameterized its geometry. Using this information, we developed the first multi-scale FEM-based model of the sensory hair and simulated a single-touch stimulus. Thereafter, we evaluated the action of the stimulus on the sensory cells in terms of two mechanotransductive parameters calculated from the cell wall stretch.

### 3.1. Representative Sensory Hair Morphology

In order to build a macro model of the entire sensory hair, we identified six characteristic sections along the hair (Figure 6) and measured the thicknesses after processing the μ-CT data in Section 2.1.1. The first section is the pointed tip of the distal lever, denoted by [$r_{1},h_{1}$]. The radius uniformly increases towards the bottom up to [$r_{2},h_{2}$], just before the onset of a bulge [$r_{3},h_{3}$] where the sensory hair has the maximum radius. Below the bulge, the constriction tissue thins down to a sharp notch [$r_{4},h_{4}$] where the radius $r_{4}$ is at its minimum, thus giving the sensory cells, their kidney-like shape. Lastly, the proximal podium constitutes the most proximal tissue with a relatively constant radius, where the top and the bottom sections are quantified by [$r_{5},h_{5}$] and [$r_{6},h_{6}$] respectively. In Table 1, the values [$r_{i},h_{i}$] for i∈ [1, 6] of all the six sections are given as a piece-wise linear function of $h^{*}$, with $k_{r}$ and $k_{h}$ being the respective multipliers.

### 3.2. Characteristic Cellular Shapes

We measured the geometric features (see Table 2) of the internal cells and reduced them to a set of parameters (see Table 3) to build micro models of the lever and podium tissues as described in Section 2.3.1. The lever consists of slender cylindrical cells (see Figure 3a) with lumen height $H_{c}=198.2\pm 46.3\mu m$ and lumen diameter $D_{c}=9.5\pm 1.1\mu m$ measured from (n = 34) cells shown in Figure 2b. In contrast, measurements from (n = 17) cylindrical cells in the podium revealed that they are significantly shorter in height, measuring $H_{c}=27.2\pm 3.8\mu m$, but they have a larger lumen diameter $D_{c}=13.9\pm 2.6\mu m$. The cell wall thicknesses in both of these tissues were similar with $T_{c}=1.1\pm 0.2\mu m$.

The constriction comprises, on the outside, of sensory cells (Figure 2c) with a characteristic kidney-like shape and, on the inside, of cylindrical cells that are similar to the cells present in the podium. There are about 30–40 sensory cells distributed evenly in a circular arrangement around the central height axis. Measurements (n = 51) taken from cross-sectional μ-CT images of the constriction revealed that the average opening angle of sensory cells is θ = 17${}^{\circ }$± 4${}^{\circ }$. The cell wall thicknesses $T_{s}$ vary along the height $H_{s}$ of the lumen and attain a maximum thickness of $T_{s,max}=2.8\pm 0.4\mu m$ at the middle of the lumen and a minimum thickness of $T_{s,min}=1.8\pm 0.5\mu m$ towards the top and bottom of the lumen. From longitudinal sectional images, geometric features of (n = 8) sensory cells were taken. The average height $H_{s}$ of the lumen is 63.3±4.8 μm and the maximum width is $D_{s,max}=26.2\pm 3.4\mu m$ and minimum width $D_{s,min}=20.8\pm 0.4\mu m$.

An important feature of the sensory cell is the thickened cell wall on the outward-facing side, which leads to the formation of a notch with a gap $L_{4}$ (Figure 2c). The cell wall is thickest above and below the notch ($L_{1,3}$), while the thinnest section is found in the middle of the notch ($L_{2}$). For the set of 8 cells, the notch lengths in each cell were normalized w.r.t. the lumen height $H_{s}$ and the corresponding ratios are: $L_{1}/H_{s}$ = 0.26 ± 0.03, $L_{2}/H_{s}$ = 0.13 ± 0.02, $L_{3}/H_{s}$ = 0.25 ± 0.05 and $L_{4}/H_{s}$ = 0.16 ± 0.03.

### 3.3. Transverse Isotropic Compliance Tensors of Hair Tissues

We homogenized the parenchymatous tissues of the sensory hair into solid models to reduce computational costs. Therefore, we calculated their transversely isotropic material properties by the procedure mentioned in Section 2.3.1, with the input parameters listed in Table 3. From the parametric study, we calculated the elastic and shear moduli and measured the Poisson’s ratios of the different tissues. The ratios of their tissue elastic moduli $E_{\|}/E_{cw}$ (longitudinal) and $E_{⊥}/E_{cw}$ (transverse) for different $E_{cw}$ and *P* are shown in Figure 7.

In Figure 7a,b, we observe that the ratios of longitudinal elastic moduli, denoted by $E_{\|}/E_{cw}$, are higher for the lever as compared to the podium. This is a consequence of the higher cell wall volume fraction $R_{v}=0.39$ of the lever compared to that of the podium with $R_{v}=0.32$. At a turgor pressure P= 0 MPa, the values of $E_{\|}/E_{cw}$ are 0.324 and 0.405 for the podium and lever, respectively. With an increase in *P* from 0 MPa to 0.3
MPa, the ratio $E_{\|}/E_{cw}$ is found to increase linearly for both tissues. At higher values of *P*, the larger fluid volume in the lumen has an additional stiffening effect. Furthermore, we observe that the rate of increase in $E_{\|}/E_{cw}$ is higher for $E_{cw}$ = 100 MPa when compared to $E_{cw}$ = 400 MPa. This shows that at lower values of $E_{cw}$, the fluid has a stronger influence on the tissue stiffness.

Interestingly, we see an opposite trend in the transverse elastic moduli $E_{⊥}/E_{cw}$, which are higher for the podium. This arises from the different cell geometries, since the lever cells are approximately six times longer than the podium cells. Hence, there is more cell wall material per unit length of the podium present in the walls between two adjacent cells along the ‖ direction. At P=0MPa, $E_{⊥}/E_{cw}$ has a lower value of 0.073 for the lever in comparison to 0.155 for the podium. For both tissues, the ratio $E_{⊥}/E_{cw}$ increases with increasing *P* similar to the longitudinal elastic moduli. Ratios of $G_{⊥\|}/E_{cw}$, $G_{⊥⊥}/E_{cw}$ and the Poisson’s ratio $\nu_{\|⊥}$ are shown in Section A.1.

### 3.4. Inverse Determination of Material Properties with Hair Stiffness

The experimental values of the moment-based stiffness *k* were determined for 7 sensory hairs via n = 17 tests. In Figure 4b, scatter plots of two experiments (blue circles) with hairs having the smallest $k_{min}=4026.0\mu N$ and largest stiffness $k_{max}=7846.5\mu N$, respectively, are shown up to a displacement of Δx=100μm. Using FEM, we mimicked the same tests wherein a force sensor contacts the hair and displaces it by Δx=300μm (Figure 4a). The scatter plot (red diamonds) shows the numerical stiffness value $k_{200}=6333.9\mu N$ for an isotropic cell wall elastic modulus $E_{cw}=200M\mathrm{Pa}$ and turgor pressure P=0.2MPa.

The same simulation was repeated for $E_{cw}=100M\mathrm{Pa}$ and 300 MPa at the same turgor pressure P=0.2MPa. In Figure 4c, the horizontal lines of the blue boxplot denote quartile values of the experimental moment-based stiffnesses. The limits of the experimental moment-based stiffness $k_{min,max}$ are marked by the top and bottom whiskers and the middle line denotes the median value k=5519.8μN. The average stiffness is k=5735.0±1282.18μN and is marked by a ‘+’ symbol. On the right, the dashed red lines correspond to the numerical moment-based stiffnesses $k_{100,200,300}$ for $E_{cw}$ = 100 MPa, 200 MPa, and 300 MPa, respectively. The simulated lower and upper limits of the stiffness ranges from 3184.7 μN to 9485.7 μN for $E_{cw}=100\;\mathrm{MPa}$ and 300 MPa, respectively. Lastly, we simulated another model with $E_{cw}=200\;\mathrm{MPa}$ but this time with P=0 MPa to test the sensitivity to the assumed values of turgor pressure in the cells. The resulting moment-based stiffness at P=0 MPa was k=6301.7 μN as compared to $k_{200}=6333.9\;\mu N$ at P=0.2 MPa.

### 3.5. Stimulus Transformation onto Sensory Cells

In the sensory hair model (see Figure 4a), the deflection deforms the hair tissues and the sensory cells embedded in them. This can change the membrane tension on the inner lumen surface of the sensory cells, which may open MSCs [12]. Therefore, we calculated the cell wall stretch $\lambda^{n}$ in each element *n* via Equation (Equation 2) located on the inner lumen surface, shown with green color in Figure 5c,d.

Cells located at ($0^{\circ }\leq \phi \leq 90^{\circ }$) are stretched while cells located ($90^{\circ }\leq \phi \leq 180^{\circ }$) undergo contraction (see Figure 5c). Figure 8a shows the stretch produced on the inner lumen surface of the cells located at $\phi =0^{\circ },90^{\circ }$, and $180^{\circ }$ at three deflection stages. Prior to deflection, i.e., $\alpha =0^{\circ }$, the stretch in the elements of the three lumina is approximately $\lambda^{n}=1$. This stretch is due to the turgor pressure P=0.2 MPa in the cell lumen. As the probe contacts the lever, it deflects the hair by an angular displacement of $\alpha =8.4^{\circ }$ and, subsequently, up to $\alpha =16.9^{\circ }$ at Δx=300 μm.

At $\alpha =16.9^{\circ }$, the cell at $\phi =0^{\circ }$ experiences a maximum stretch of $\lambda^{n}=1.036$, whereas the cell located diametrically opposite at $\phi =180^{\circ }$ undergoes contraction resulting in $\lambda^{n}=0.965$. The highly stretched and contracted regions in the sensory cells can be spotted as red and blue zones, respectively, in Figure 8a.

#### 3.5.1. Coarse-Grained Properties in Sensory Cells

Coarse-grained properties are calculated to compare the deformation among the different cells present in the model. The first property is the cumulative area-normalized stretch $\delta_{\phi }$ (see Equation (Equation 3)), located at polar angle ϕ. Figure 8b shows the relative change in cumulative stretch $\Delta \delta_{\phi }$ w.r.t. the pre-deflection stage at $\alpha =0^{\circ }$ for two cases: with fluid (diamonds) and without fluid (circles) in the cell lumina. We made a comparison between the two cases to investigate the effect of turgor pressure on cell wall stretch. When the hair with fluid-filled cells is deflected from $\alpha =0^{\circ }$ to $16.9^{\circ }$, $\Delta \delta_{0}$ increases from 0 to 0.0079 due to stretch, while $\Delta \delta_{180}$ attains a negative value of −0.0086 due to contraction. An important observation is that, when there is no fluid in the cells, $\Delta \delta_{0}$ increases more rapidly (linear slope = $1.02\times 10^{-3}$) than with fluid (linear slope = $4.67\times 10^{-4}$). Therefore, the stretch and contraction of the sensory cells at $\alpha =16.9^{\circ }$ is almost twice as large in empty cells in comparison to fluid-filled cells.

Figure 8c shows the change in *P* in the cell lumen caused by the deformation of the cells. At the onset of deflection, all the 16 cells have the same initial P=0.2 MPa marked by blue circles. As the hair is deflected to α = $8.4^{\circ }$ and further up to $16.9^{\circ }$, shown in red and yellow circles, respectively, *P* varies inside the 16 cells. *P* attains values between −0.66
MPa and 1.2
MPa in the cells located between $\phi =0^{\circ }$ and $\phi =180^{\circ }$ due to the tension and compression in the respective cells. The wide range of values attained by *P* is due to the assumption that there is no exchange of the incompressible fluid during stimulation.

#### 3.5.2. Influence of Cellular Shape on the Stretch Pattern

In Figure 8a, we observe that the cell wall is highly stretched (red zones) and contracted (blue zones) in the top and bottom regions of the original cell (O.C). To quantify these differences, we divided the lumen into three regions: top, middle, and bottom (see Figure 9a). Then, we calculated the regional cumulative stretch $\delta_{top,mid,bot}$ and the sensitive area fraction $\rho_{top,mid,bot}$ using Equations (Equation 3) and (Equation 4) in the O.C and in the two additional cell shapes, the notchless cell (N.C) and the cell with uniform wall thickness (U.C). By comparing these values among the shape variants, we demonstrate how cell morphology influences the cell wall stretch.

We find that, in the case of the N.C and O.C, the top and bottom regions undergo the highest stretch and contraction at $\phi =0^{\circ }$ and $\phi =180^{\circ }$, respectively (squares and circles, see Figure 9b). This is because the cell wall is thicker in the middle as compared to the other regions. $\delta_{top,bot}$ for O.C and N.C do not vary significantly; however, we see a higher stretch $\delta_{mid}=1.013$ for N.C (diamonds, Figure 9b) compared to $\delta_{mid}=1.008$ for O.C. This is due to the fact that, when we artificially eliminated the notch in the N.C, the total cell wall volume is reduced in comparison to the O.C. Consequently, the middle region of the N.C undergoes a higher stretch due to a lower stiffening effect. Interestingly, we see a complete reversal of trends when we compare the O.C with the U.C and find that highest stretch and contraction is found in the middle region for $\phi =0^{\circ }$ and $\phi =180^{\circ }$, respectively. This is seen in Figure 9b where $\delta_{mid}$ (diamonds) attains a maximum of 1.014 and a minimum of 0.988 in the case of the U.C.

We calculated $\rho_{top,mid,bot}$ based on the assumption that MSCs open when a threshold stretch or contraction is produced. Our intent was to compare the sensitivity of the sub-regions with respect to each other. To this end, we selected a 2% change in stretch from the reference value of $\lambda^{n}=1$ as our threshold, which translates to $\lambda_{max}^{n}=1.02$ and $\lambda_{min}^{n}=0.98$. In Figure 9c (left), we find that for the O.C and the N.C approximately 40% of the top and bottom regions (squares and circles) is sensitive, while less than 2% in the middle region is sensitive $\rho_{mid}\leq 0.02$. On the other hand, when we evaluate the sensitivity of the sub-regions in the U.C shape variant, it is the middle region (diamonds) that is more sensitive. $\rho_{mid}=$ attains a maximum of 0.45 and a minimum of 0.27 for the cells at $\phi =0^{\circ }$ and $\phi =180^{\circ }$, respectively.

## 4. Discussion

The different aspects of mechanotransduction in the *Venus flytrap* were investigated through various theoretical and experimental approaches. Previously, we introduced an electromechanical model [10] built on the concepts of plant electrical memory [4] to complement our force-deflection tests on sensory hairs. We found hair deflection thresholds for angular displacement and velocity which successfully elicit action potentials, thereby closing the trap. Continuing this work, we hypothesized that, at the cellular level, there could be mechanotransductive properties whose thresholds should relate to the sensory hair’s deflection. Hence, we were interested in studying the effect of the mechanical stimulus on the sensory cells located at the constriction near the base of the sensory hair. We pursued this topic from a computational perspective to overcome the experimental challenges and were able to evaluate how the sensory cells deform as a function of hair deflection.

We focused on the internal cellular morphology of the sensory cells and studied its influence on the kinematics behind mechanotransduction in the sensory hair. Prior to this, researchers have investigated the hair’s internal structure [7,8,22]; however, these studies were based on microscopic observations and 2D TEM images, which are not sufficient to build a comprehensive 3D multi-scale model of the sensory hair. Therefore, we acquired μ-CT data of the hairs and parameterized the relevant geometric features needed to build our models. In this process, we quantified the morphology of the sensory hair and its cells. Other important measurements obtained from the μ-CT data are the cell wall thickness of the peripheral cells of the hair and the precise geometry of the notch.

With these morphometric data, we built a multi-scale model of the sensory hair. To our knowledge, this is the first multi-scale 3D model of the hair, which we used to simulate single-touch stimuli. We built micro models of the internal tissues to investigate the effect of turgor pressure *P* and isotropic cell wall elastic moduli $E_{cw}$ on the overall hair stiffness. From a series of simulations, we observed that $E_{cw}$ is the dominant parameter affecting the overall material behavior as changing from P=0.2 MPa to P=0 MPa led to a minimal change in $M_{x}$ (less than 1%). Then, we were able to approximate the orthotropic material properties by comparing the moment vs. deflection curves obtained via experiments as well as from those obtained through simulations. We found that the combination of $E_{cw}=200\;\mathrm{MPa}$ and P=0.2 MPa best represents the sensory hair’s material. The numerical curves appear to be smoother than the experimental ones, because the outer surface of the real hair contains fine grooves oriented in the ‖ direction (seen via Scanning Electron Microscopy [22]), which are not included in the model. These grooves cause the force probe to slip and jump during the experiments, leading to the artifacts seen in the curves.

Thereafter, we simulated the hair deflection to mimic a natural stimulus for two additional artificial cell shapes. The first comparison was made between the original cell (O.C) having a sharp notch and the notchless cell (N.C), both with varying wall thicknesses. Upon comparing both, we found that the notch does not severely affect the cell wall stretch $\delta_{\phi }$. Hence, we can infer that the notch may not play a direct role in mechanotransduction. However, the notch is likely a protective measure developed by the plant. When the hair is excessively deflected, as it is the case in a closed flytrap, the two ends of the notch contact each other. In this way, only the lever tissue above the notch deforms, thus preventing any damage to the sensory cells.

Finally, we designed a third cell shape variant with uniform cell wall thickness (U.C) and compared the mechanotransductive properties with the O.C. We observed that there is a reversal of trends with regard to the location of the highly stretched regions as well as the sensitivity of the sub-regions. We found that, in the O.C, the top and bottom regions are highly sensitive but, in the U.C, it is the middle region, which experiences the largest stretch and contraction. This is a consequence of the thicker cell walls in the middle of the O.C, restricting large deformations. This finding suggests that the morphology of the O.C favors the localization of higher stretch in smaller zones rather than a uniform stretch across the entire inner lumen surface. There are two possible explanations for this behavior. Firstly, it may be the case that only a small number of open MSCs are sufficient for mechanotransduction in the sensory cells. Alternately, the MSCs may largely be concentrated in specific regions to form ‘hotspots’, with higher sensitivity. Our findings concerning the MSCs in the flytrap could be of interest to electrophysiolgists studying similar species as well as for the investigation of plant specimens in general.

Through our model based on novel μ-CT scans, we were able to capture the effects of the geometric non-linearity arising from the characteristic morphological features of the internal cells and their internal fluid. This numerical framework can be expanded to include material non-linearity to further investigate the rate-dependent behavior of the flytrap’s electrical response when stimulated at different velocities. Another extension of this work could be to include fluid exchange in the tissues and analyze the competing effects of material relaxation and changing fluid volume within the sensory cells. Additionally, comparisons can be made with images of sensory cells captured while the hair is bent. However, alternate imaging techniques may be used to scan the hair in the bent state as the current μ-CT equipment turned out to be unfeasible. This is because, the required exposure times were not short enough to avoid substantial radiation damage. On a broader scope, our analysis may be beneficial for the biosensor industry, which is regularly inspired by sensing mechanisms found in nature. Inspiring structural features can be found hidden amongst the salient features of the sensory hair’s geometry, for instance the presence of a constriction, the characteristic shape of the sensory cells, the lever arm arrangement, or the positioning of the sensory cells along the hair’s periphery. Future work could be directed at integrating these features into the development of bio-inspired sensors.

## Figures and Tables

**Figure 1 ijms-22-00280-f001:**
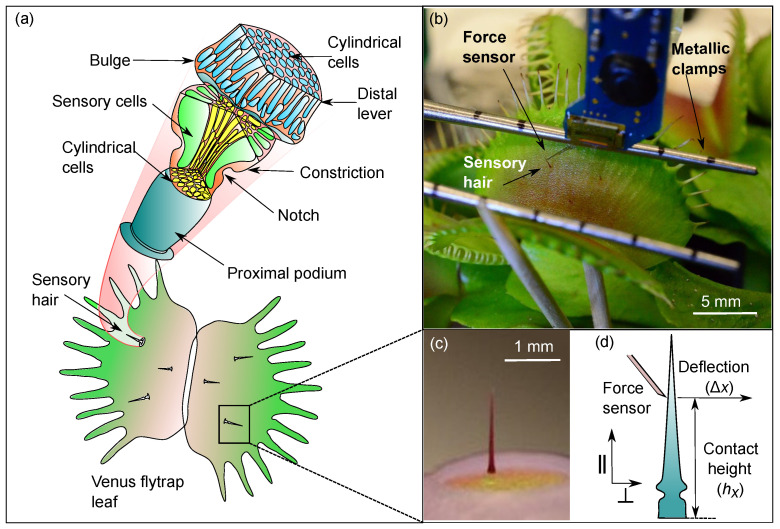
(**a**) Anatomical diagram of the sensory hair showing the various tissues (podium, constriction, lever) and cells (sensory, cylindrical). (**b**) Experimental set up of the hair deflection tests comprising of a MEMS-based force sensor to impart stimuli and metallic clamps to constrain lobe movement. (**c**) A sensory hair taken from the flytrap lobe prepared for μ-CT imaging. (**d**) Force sensor contacting the hair at a height $h_{x}$ from the upper lobe epidermis, deflecting it horizontally by Δx.

**Figure 2 ijms-22-00280-f002:**
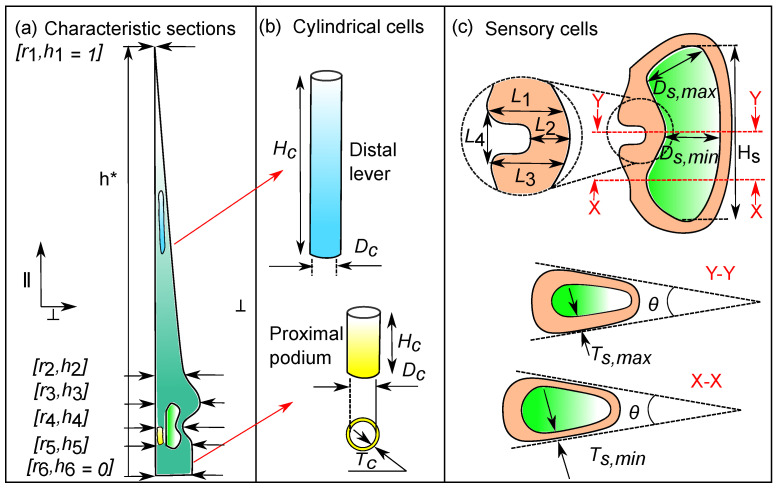
(**a**) Characteristic sections of the hair geometry denoted by normalized height $h_{i}$ and radius $r_{i}$. The ‖ and ⊥ arrows correspond to the longitudinal and transverse directions. (**b**) Cylindrical cells in the lever (**blue**) and podium (**yellow**) with diameters $D_{c}$, heights $H_{c}$, and cell wall thickness $T_{c}$. (**c**) Longitudinal section (**top**) of sensory cell with the inner lumen (**green**) having height $H_{s}$ and widths $D_{s,max}$ and $D_{s,min}$. A magnified view of the notch is shown with features $L_{1,2,3,4}$. The cross-sections (**bottom**), X-X and Y-Y, depict the cell’s opening angle θ and the wall thicknesses $T_{s,min}$ and $T_{s,max}$.

**Figure 3 ijms-22-00280-f003:**
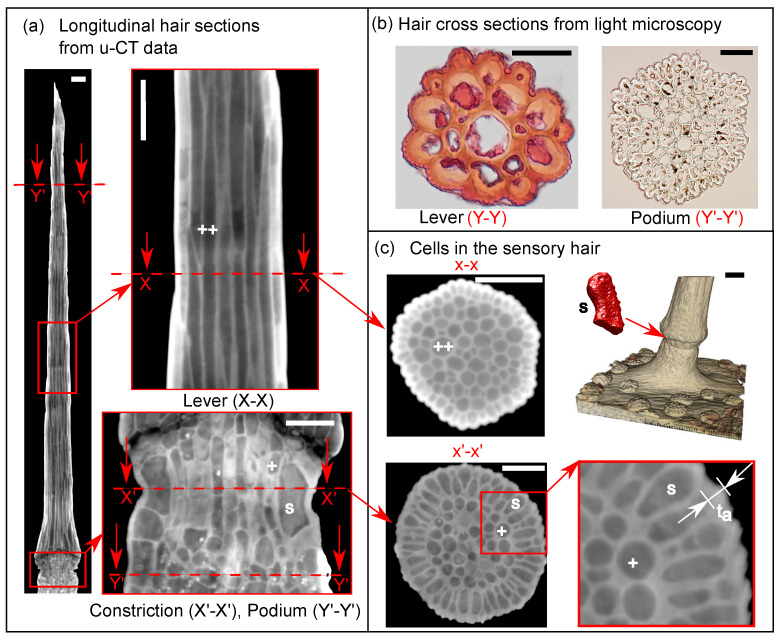
(**a**) Longitudinal μ− CT section of sensory hair with enlarged views (**right**) of the hair’s tissues. Sensory cells ‘s’ and cylindrical cells ‘+’, ‘++’ are seen in the podium (**right-bottom**) and lever (**right-top**), respectively. Scalebar = 50 μm (**b**) Light microscopy images of lever cross section stained with Safranine solution (Y-Y, left) and podium (Y’-Y’, right), excised at regions marked in the longitudinal sections. Scalebar = 20 μm (**c**) μ−CT images of lever (top-left, X-X) and podium (bottom-left, X’-X’) cross sections. Computer rendering (**top-right**) of the sensory hair and the sensory cell (red) from μ-CT data. An enlarged view of the cross section (X’-X’) shows the thickened cell wall $t_{a}$. Scalebar = 50 μm.

**Figure 4 ijms-22-00280-f004:**
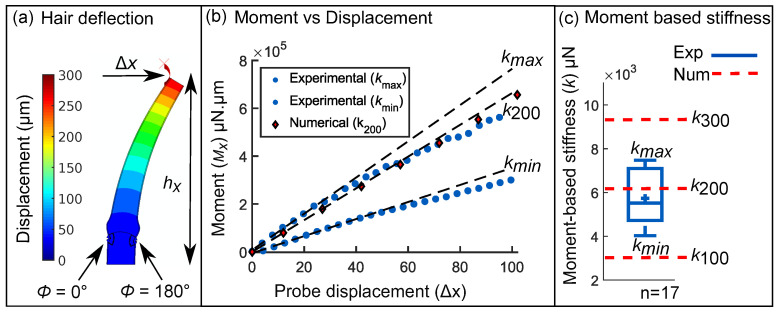
(**a**) FEM simulation output showing the deflection of a sensory hair using a displacement up to Δx=300 μm. (**b**) Moment vs. displacement curves (experimental, blue circles) shown for two force-displacement tests and (numerical, red diamonds) for $E_{cw}$ = 200 MPa. The initial linear slopes $k_{max,min}$ give the lowest and highest moment-based stiffness values found in (n = 17) tests. (**c**) Boxplot of the moment vs. displacement slopes *k* obtained from 17 experimental tests. The middle line denotes the median value and the mean is shown with a ‘+’ sign. Numerical outputs are denoted by dashed red lines for $E_{cw}$ = 100, 200, and 300 MPa.

**Figure 5 ijms-22-00280-f005:**
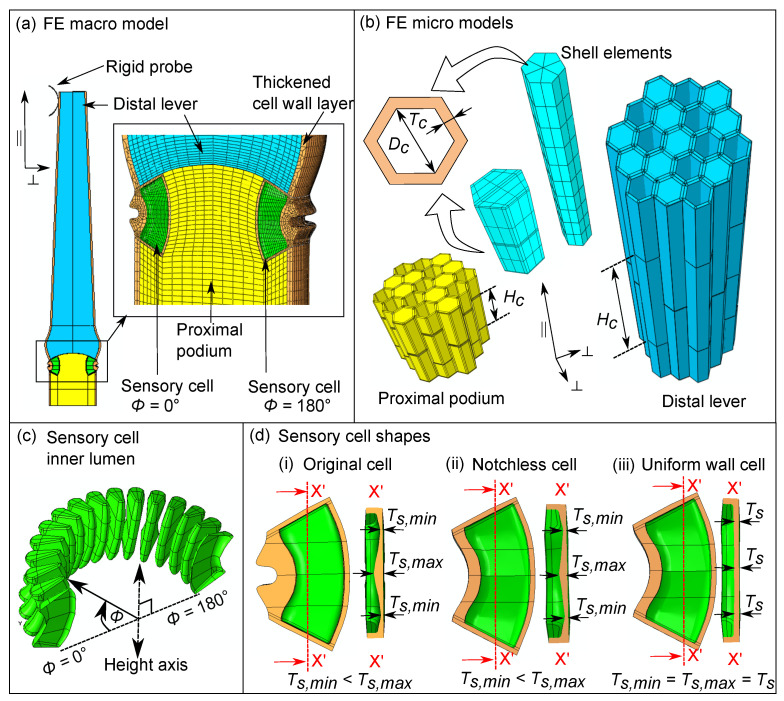
(**a**) Sensory hair half model with lever (**blue**), podium (**yellow**), and sensory cells with their inner lumen (**green**) and cell wall (**beige**). The enlarged image (**right**) shows FEM mesh. (**b**) Tissue micro models with hexagonal cells having a height ($H_{c}$), lumen diameter ($D_{c}$), and wall thickness ($T_{c}$). (**c**) Segmented view of the inner lumina of sensory cells located at polar angle ϕ∈ [0, 180], about the height axis. (**d**) Longitudinal section of sensory cells with (**i**) original cell shape and two variants: (**ii**) a notchless cell and (**iii**) a sensory cell with uniform wall thickness. $T_{s,min,max}$ denote cell wall thicknesses at different regions.

**Figure 6 ijms-22-00280-f006:**
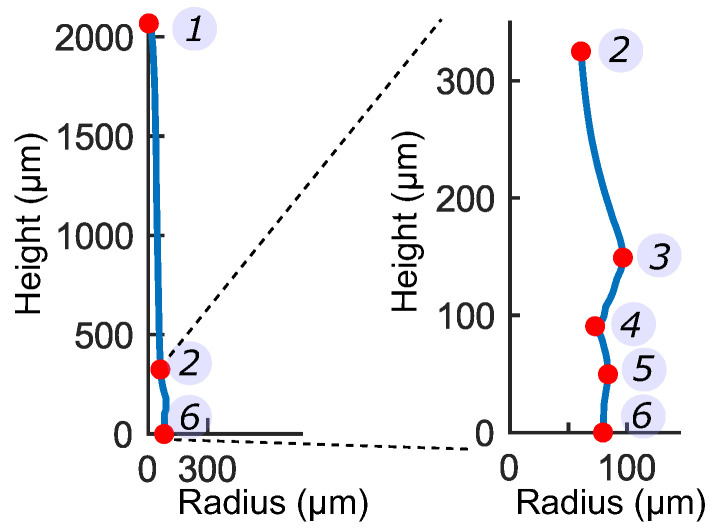
Characteristic Sections (1–6, red dots) are identified on one sensory hair. Magnified view of the hair base shown on the right.

**Figure 7 ijms-22-00280-f007:**
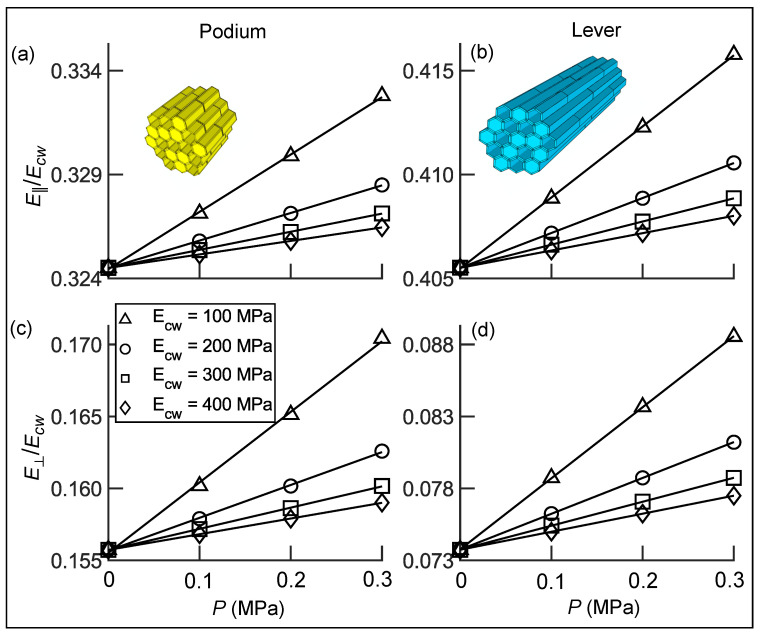
(**a**,**b**) Ratio of longitudinal elastic moduli $E_{\|}/E_{cw}$ of the podium and the lever respectively. (**c**,**d**) Ratio of the transverse elastic moduli $E_{⊥}/E_{cw}$ of the podium and the lever respectively, calculated for different turgor pressures *P* and cell wall elastic moduli $E_{cw}$ (∆ =100MPa, ○=200MPa, ☐=300MPa and ⋄=400MPa).

**Figure 8 ijms-22-00280-f008:**
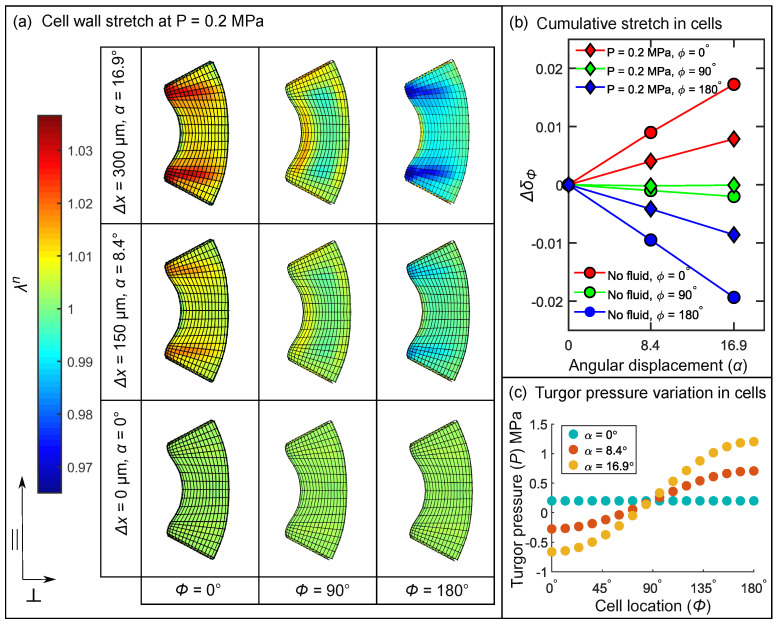
(**a**) Stretch $\lambda^{n}$ in the elements of the sensory cells located at ($\phi =0^{\circ },90^{\circ },180^{\circ }$) shown for 3 angular displacements α. (**b**) Change in cumulative area-normalized stretch $\Delta \delta_{\phi }$ for cells located at $\phi =0^{\circ }$ (**red**), $\phi =90^{\circ }$ (**green**) and $\phi =180^{\circ }$ (**blue**) with increasing angular displacement α for cells with fluid (**diamonds**) and without fluid (**circles**). (**c**) Variation of fluid cavity pressure *P* in the 16 sensory cells at 3 stages of deflection $\alpha =0^{\circ }$ (**blue**), $\alpha =8.4^{\circ }$ (**red**), and $16.9^{\circ }$ (**yellow**).

**Figure 9 ijms-22-00280-f009:**
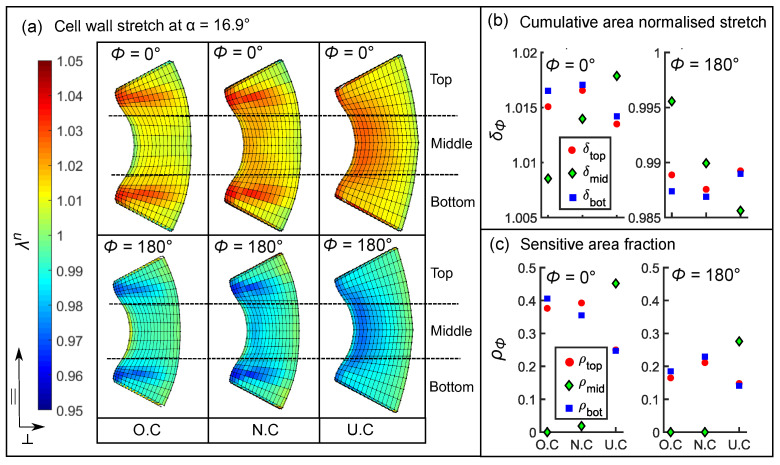
(**a**) Stretch $\lambda^{n}$ in the inner lumen surface of the two sensory cells located diametrically opposite each other at $\phi =0^{\circ }$ (**top**) and $\phi =180^{\circ }$ (**bottom**) at an angular displacement ($\alpha =16.9^{\circ }$), shown for the original cell (O.C), notchless cell (N.C) and the cell with uniform wall thickness (U.C). (**b**) Cumulative area-normalized stretch $\delta_{\phi }$ plotted for the O.C, N.C, and U.C at $\alpha =16.9^{\circ }$, in the top, middle, and bottom sub-regions on the inner lumen surface. (**c**) Sensitive area fraction $\rho_{\phi }$ plotted for the sub-regions in the three cell shape types at $\alpha =16.9^{\circ }$.

**Table 1 ijms-22-00280-t001:** Normalized heights $h_{i}$ and radii $r_{i}$ of characteristic sections, averaged for three sensory hairs, are expressed as linear functions: $r_{i}=k_{r}\;\cdot \;h^{*}$ and $h_{i}=k_{h}\;\cdot \;h^{*}$, where $h^{*}$ is the normalized total hair height.

Tissue	Section	$k_{r}$	$k_{h}$
Distal lever (top)	[$r_{1},h_{1}$]	0	1
Distal lever (bottom)	[$r_{2},h_{2}$]	0.032	0.174
Bulge	[$r_{3},h_{3}$]	0.050	0.079
Notch	[$r_{4},h_{4}$]	0.041	0.049
Proximal podium (top)	[$r_{5},h_{5}$]	0.045	0.029
Proximal podium (bottom)	[$r_{6},h_{6}$]	0.045	0

**Table 2 ijms-22-00280-t002:** Characteristic features of different cells.

Cell Type	Height (μm)	Diameter (μm)	Wall Thickness (μm)
Cylindrical cells (lever)	$H_{c}$ = 198.2 ± 46.3	$D_{c}$ = 9.5 ± 1.1	$T_{c}$ = 1.1 ± 0.2
Cylindrical cells (podium)	$H_{c}$ = 27.2 ± 3.8	$D_{c}$ = 13.9 ± 2.6	$T_{c}$ = 1.1 ± 0.2
Sensory cells	$H_{s}$ = 63.3 ± 4.8	$D_{s,max}$ = 26.2 ± 3.4, $D_{s,min}$ = 20.8 ± 2.5	$T_{s,max}$ = 2.8 ± 0.4, $T_{s,min}$ = 1.8 ± 0.5

**Table 3 ijms-22-00280-t003:** Representative cellular features in the Finite Element Method (FEM) micro models.

	Proximal Podium	Distal Lever
Cell height $H_{c}$ [μm]	30	200
Cell lumen diameter $D_{c}$ [μm]	14	10
Cell wall thickness $T_{c}$ [μm]	1	1
Cell wall Young’s modulus $E_{cw}$ [MPa]	100–400	
Cell wall Poisson’s number $\nu_{cw}$ [-]	0.3	
Turgor pressure *P* [MPa]	0–0.3	
Cell wall volume fraction $R_{v}$ [-]	0.32	0.39

## Data Availability

Data available in a publicly accessible repository.

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
