# Peer review of "Kinematics Governing Mechanotransduction in the Sensory Hair of the Venus flytrap"

_ijms, 2020, doi:10.3390/ijms22010280_

Round 1
Reviewer 1 Report
The authors propose a study titled "Kinematics governing mechanotransduction in the sensory hair of the Venus flytrap". The work is original, well written and modulated in the correct way, and worthy to published.
I suggested to complete the discussion with two three rows that indicate its possible application for industrial purposes?, to control harmful insects? in the agriculture fields? or other...
Other few correction highlighted in green in the text

Author Response
We are delighted to receive your feedback and have incorporated a few sentences at the end of the manuscript as you have suggested. Additionally, we confirmed the name of the microscope used during our micro-CT experiments from the manufacturer’s catalogue, and its name is spelled as: pco.edge 5.5. Accordingly, we have made this change.
Reviewer 2 Report
I have carefully read the manuscript “Kinematics governing mechanotransduction in the sensory hair of the Venus flytrap” submitted by Saikia and colleagues. This is a very well written and scientifically sound piece of scientific work. My sincere congratulations to the authors for this highly interesting study!
Besides few minor points listed below, my only main criticism is that the authors have not looked at a bent sensory hair via µCT to back up their theoretical results gained from modelling regarding the bending deformation. I believe that it is feasible to “freeze” a trigger hair in the bent state, e.g. via clamps or glue, and then to look at the cellular deformation. Such a direct comparison would greatly up-value the results and discussion presented in the manuscript.
Minor points:
I wonder whether the preparation of trigger hairs and the µCT scanning process cause drying artefacts? Why were the hairs not stored in a small chamber with a humid environment?
The two lobes of the trap were constrained with clamps to stay open. Did this rough mechanical perturbation not cause any closing response? See “Touch Receptor of Venus Flytrap, Dionaea muscipula” by DiPalma et al. 1966 Science.
Author Response
Dear Reviewer
Please find the attached PDF file for our responses
Best
Authors
